# Relationships among thriving at work, organisational commitment and job satisfaction among Chinese front-line primary public health workers during COVID-19 pandemic: a structural equation model analysis

Mo Yi ,[1] Di Jiang,[2] Jingjing Wang,[1] Zeyi Zhang,[1] Yuanmin Jia ,[1] Baosheng Zhao,[1] Lei Guo,[3,4] Ou Chen[1]

MY and DJ are joint first authors.

For numbered affiliations see end of article.

**Correspondence to**
Prof Ou Chen;
chenou@sdu.edu.cn

## ABSTRACT

**Objectives** This study aims to explore the mediating effect and influence mechanism of organisational commitment on the association among thriving at work and job satisfaction among frontline primary public health workers (PHWs) in China during the COVID-19 pandemic.

**Design** This study is a cross-sectional written survey.

**Setting** We included 20 primary care units in northern provinces of China.

**Participants** A total of 601 PHWs who worked in primary organisations and against COVID-19 on the front line were included.

**Methods** We collected the data from the participants' written questionnaire (Minnesota Satisfaction Questionnaire, thriving at work scale and organisational commitment scale), and programmed AMOS V.26.0 to develop a structural equation model (SEM) based on the relationships among the three variables.

**Results** The thriving at work scores of the primary PHWs were (M=3.17, SD=0.65), and job satisfaction was (M=3.05, SD=0.69); the scores of their thriving at work, organisational commitment and job satisfaction were all significantly correlated (p<0.01); and the SEM indicated that organisational commitment had a significant partial mediating effect between thriving at work and job satisfaction. The overall effect value was 0.867, and the mediated effect value was 0.422, accounting for 48.7% of the total effect size.

**Conclusion** The thriving at work and job satisfaction scores of primary PHWs in China are moderate, and thriving at work not only affects job satisfaction directly, but also indirectly through organisational commitment. This study suggests that health policy-makers should promote job satisfaction among PHWs through relative inventions aiming to improve their thriving at work and organisational commitment.

## STRENGTHS AND LIMITATIONS OF THIS STUDY

⇒ The structural equation model was used to analyse the associations among thriving at work, organisational commitment and job satisfaction.

⇒ The covariates, including participants' gender, age, years of working and professional level, were adjusted to ensure the accuracy of the statistical analysis.

⇒ Only primary public health workers (PHWs) in the eastern region of China were included in the study.

⇒ The cross-sectional nature of this study cannot be used to deduce the causal relationships among the three variables.

⇒ This study focused more specifically on the frontline pandemic-fighting population of primary PHWs.

millions of infections and deaths worldwide. The WHO identified the outbreak as a Public Health Emergency of International Concern, which poses a huge threat to people's health and the global economy.[1 2] As lower-level medical organisations, China's primary healthcare units played an important role in the battle against COVID-19 on the front line of defence.[3]

Public health is a significant part of China's healthcare service system and an indispensable force in improving the health of community residents. Primary public health workers (PHWs) are responsible for the detection, prevention and control of various infectious diseases, occupational health examinations, planned immunisations and epidemiological surveys.[4 5] In China, PHWs are a separate group of healthcare providers, different from other professional groups, such as physicians, pharmacists and nurses. They are mainly responsible for epidemiological

## INTRODUCTION
In early 2020, an epidemic called COVID-19 suddenly swept the world, ultimately causing

investigations, quarantine management of patients with infectious diseases, environmental disinfections and specimen collections in the community or other primary organisations. As the essential gatekeepers in epidemic control, since the outbreak, primary PHWs have not only completed heavy workloads, but also risked being infected while working on the front line.[6 7] They are in a state of chronic psychological stress and gradually experience emotional exhaustion from their profession. In addition, their personal physical, energetic and motivational strengths are in a state of increasing overload, which has negative effects on their job satisfaction and thriving at work.[8]

Job satisfaction is the subjective psychological state of employees' physiological and psychological perceptions of the work environment and the work itself.[9] Veličković's empirical study subdivided job satisfaction into internal and external satisfaction.[10] Currently, job satisfaction is used as the best predictor to measure employees' self-fulfilment of their professional values.[5 11] Unlike the development of large hospitals, primary care units in China are faced with multiple challenges, such as smaller platforms with limited opportunities, employees' lower salaries and weaker medical service delivery capacities.[12] PHWs have not received much attention for a long time, compared with physicians and nurses. This is why our study focuses on the current status of job satisfaction and, thriving at work of primary PHWs with their occupation and its environment.

Thriving at work is a positive psychological state characterised by a joint sense of vitality and learning, that ultimately contributed to the flourishing development of the organisation according to Spreitzer.[13] A socially embedded model of thriving at work suggests that vitality represents the hedonic component of well-being, whereas learning, is a means of realising one's potential, which indicates job satisfaction is an external display of thriving at work.[14] Thriving at work is a crucial factor influencing healthcare providers' personal career development and planning, accordingly contributing to their professional fulfilment and maintaining health.[15] Zhao et al's study pointed out that healthcare providers in China who lacked thriving at work often experienced low job satisfaction, which affected their work enthusiasm and stability.[16] This indicates a highly positive correlation between job satisfaction and thriving at work, both of which are directly related to the level of quality of healthcare services.[5 17]

Organisational commitment usually refers to the relative strength of an individual's identification with and involvement in a particular organisation.[18] Meyer and Allen argued that organisational commitment is a psychological phenomenon that encourages employees to stay with the organisation and participate wholeheartedly in its work as their 'side-bet'; they regard it as consisting of three components: affective, continuance and normative commitment.[19] The level of organisational commitment increases from intrinsic value, which is another key expression of thriving at work. A recent study by Kleine confirmed a positive relationship between thriving at work and organisational commitment (RC=0.63).[20] A study that concentrated on nurses noted that organisational commitment was a significant positive predictor of their job satisfaction.[21] In addition, researchers confirm that each component of organisational commitment is a critical determinant that affects an individual's job satisfaction and life satisfaction.[22 23]

In the field of occupational research and medical practice, the job demands-resources (JD-R) model proposed by Bakker recently attracted attention with regard to creating organisational culture, improving employee job satisfaction and reducing turnover during the pandemic.[24] Job demands usually refer to the extent to which the duties of the job require employees to exert their own somatic, social or psychological (cognitive and emotional) efforts, such as workload, emotional demands and hostile work environment. Job resources, on the other hand, consist of a set of components that are beneficial or supportive to the employee, mainly in terms of intrinsic motivation, social praise or organisational belonging.[25] The JD-R model has been tested and validated in a large sample of different countries and industries.[26–28] We consider the increased workload and physical and mental exhaustion of primary PHWs during an epidemic as high job demands, and employees' organisational commitment and perceived sense of thriving as job resources. Job demands and job resources interact mutually and are reflected in the job satisfaction which is the main outcome. Therefore, the model needs to be used to further explore what factors of JD-R are associated with job satisfaction.

It is known that job satisfaction, thriving at work and organisational commitment are the key factors that affect employee job stability and that the individual has a stable job for a certain period of time and does not have the idea of leaving easily.[29] However, the inherent influence mechanism among the three variables of PHWs is still unclear. Most existing studies have explored the job satisfaction of physicians or nurses in large hospitals, with less attention given to the public health personnel group. Therefore, this study is equipped with some scientific research value for human resources in the field of PHWs. Therefore, the first aim of this study was to assess the current status of thriving at work and job satisfaction among primary PHWs during the COVID-19 pandemic. The second aim was to construct a structural equation model (SEM) for these three variables among primary PHWs, empirically analyse their quantitative relationships and provide suggestions on the relevant policies and healthcare practices, because it is crucial to stabilise the workforce of primary PHWs by improving their job satisfaction. Meanwhile, we hypothesised that (1) thriving at work and organisational commitment are protective factors for job satisfaction among primary PHWs and (2) thriving at work indirectly and directly affects job satisfaction through organisational commitment.

## METHODS

### Study participants

This study was a cross-sectional study. A convenience sampling method was used to determine the 20 primary healthcare units (community/township health service centres, rural clinics) in northern provinces of China with whom we had a partnership, and the random number table was used to determine the targeting of 650 primary PHWs from May to October 2020.

The inclusion criteria for participants were as follows: (1) years of work in primary care units≥1 year; (2) having participated in front-line work against the COVID-19 epidemic for more than 1 month and (3) professional qualification as a licenced registered PHW. The exclusion criteria were as follows: (1) interns and trainees; (2) those whose work sites were nonprimary organisations; and (3) physicians or nurses. A total of 650 questionnaires were distributed in this survey, and 601 valid questionnaires were finally returned, for a valid return rate of 92.46%.

### Measures

#### Sociodemographic information questionnaire

After referring to previous literature and expert consultation, we identified the following demographic information as the study variables: gender, age, education background, type of working unit (community/township healthcare service centres, rural clinics), years of working and professional level of the participants.

#### Minnesota Satisfaction Questionnaire

The Minnesota Satisfaction Questionnaire (MSQ), a short-form scale developed by Weiss, Dawis, England and Lofquist to measure employee job satisfaction, contains two dimensions: internal and external satisfaction.[30] The MSQ has 20 items and is scored on a 5-point Likert scale, with 'very dissatisfied', 'dissatisfied', 'fair', 'satisfied', 'very satisfied' and 'satisfied'. The higher the score is, the higher the satisfaction level of primary PHWs with their work. Previous studies focused on healthcare providers have shown that the Cronbach's alpha value of this scale is 0.87–0.92.[31 32]

#### Organisational commitment questionnaire

The Organisational Commitment Questionnaire (QCQ), developed by Allen and Meyer,[33] was used in this study to measure the organisational commitment of primary PHWs. It contains three dimensions: affective commitment, normative commitment and sustaining commitment, and each dimension includes six items. All items are measured on a Likert 5-point scale from 1 (strongly disagree) to 5 (strongly agree). The Cronbach's alpha values for the dimensions are 0.928, 0.802 and 0.859,[34] respectively. The total scale score was the sum of the items and ranged from 18 to 90, with higher scores representing higher levels of employee organisational commitment.[35]

#### Thriving at work scale

The thriving at work scale developed by Porath et al[36] was used; it has 10 items and measures two dimensions of thriving at work, vitality and learning, with each containing five items. This is a Likert 5-point scale from '1 to 5' representing 'very unlikely ~very likely', respectively. The higher the score is, the higher the sense of thriving at work of the participants. The Chinese version of the questionnaire has been shown to have good reliability and validity in the Chinese teacher and nurse populations.[37 38]

### Data collection

This study was an anonymous and written survey, and all the data obtained were promised confidentiality. Rigorous training and assessment of the investigators were conducted before the survey began, and the participants all signed an informed consent prior to the study. The paper questionnaires were distributed together by the investigators in cooperation with the leaders of the units or organisations, and the participants received instructions. The questionnaires were carefully checked by the investigators after the participants had completed them and were collected in time.

### Statistical analysis

SPSS V.21.0 was used to analyse the data descriptively. A t-test and one-way analysis of variance (ANOVA) were used to compare the differences among variables on the sociodemographic characteristics of the study population. Pearson correlation analysis was used to clarify the correlations among organisational commitment, job satisfaction and thriving at work among primary PHWs, with a p value smaller than 0.05 considered a statistically significant difference. Based on the correlation analysis, AMOS V.26.0 was used to develop an SEM to analyse the inherent relationships among the three variables. The path coefficients between the latent variables were tested using path analysis, and the parameter estimation was performed using the great likelihood method, assuming that the significance of the mediating effect of organisational commitment in the model was calculated using bootstrap analysis with 2000 samples with the test level set at $\alpha=0.05$, and if the 95% CI of the standardised path coefficients did not contain 0, the mediating effect was significant. The SEM can be considered well-fitted when all path coefficients satisfy the criteria of Hu and Bentler's study.[39]

## RESULTS

### General information

A total of 650 primary PHWs who had participated on the front line of epidemic fighting were surveyed in 2020. The average age of the participants was 36.61±8.95 years old, with the majority being female (82.9%). The education background of all participants was mainly college or above (94.0%). The average working years was 14.31±6.21 years, and their professional levels, which is a comprehensive reflection of their work experience and work ability, were mainly junior (72.3%). For the three main scales involved in this study, we performed the relevant Cronbach coefficient tests, and the results were as follows: (1) the Cronbach's alpha value of the MSQ was

0.919; (2) the Cronbach's alpha value of the QCQ in this study was 0.917; (3) the overall Cronbach's alpha value of the thriving at work scale was 0.89, and the Cronbach's alphas for the vitality and learning dimensions were 0.87 and 0.89, respectively.

## Analysis of common method bias test

In this study, a common method bias test was conducted using Harman's one-way test and exploratory factor analysis was performed without rotation. The results showed that there were 14 factors with characteristic roots greater than 1. The variance rate explained by the first factor of job satisfaction was 36.73%, which was less than the critical value of 40%, indicating that there was no serious problem of common method bias in this study.[40]

## Demographic differences in thriving at work and job satisfaction scores

The results showed that for primary PHWs, there were significant differences in thriving at work on two demographic variables: education and years of working. The PHWs who had obtained a lower education level had higher thriving at work; additionally, the lowest level of thriving at work was observed in the group with more than 20 years of working.

There were significant differences in job satisfaction among primary PHWs on three demographic variables: gender, education and years of working. Female PHWs had higher scores than males in job satisfaction; PHWs with more years of working had a higher level of job satisfaction; and PHWs with a better education background had lower job satisfaction, as detailed in table 1.

## Correlations among organisational commitment, job satisfaction and thriving at work among primary PHWs

The results showed that the mean score of thriving at work of the 601 responding primary PHWs was 3.17±0.65. The mean score of their organisational commitment was 2.93±0.73. The mean score of their job satisfaction was 3.05±0.69. Pearson's correlation analysis revealed that thriving at work was positively correlated with organisational commitment (r=0.715, p<0.001) and highly positively correlated with job satisfaction ($r$=0.804, p<0.001); furthermore, there was a highly positive correlation between organisational commitment and job satisfaction (r=0.848, p<0.001) (see table 2).

## SEM of organizational commitment, job satisfaction and thriving at work among primary PHWs

To investigate the influence mechanism among the variables in depth, we controlled for the gender, age, education background and years of working of the participants. An SEM was constructed with thriving at work as the independent variable, organisational commitment as

**Table 1** Demographic differences in thriving at work and job satisfaction among primary PHWs

| Variables | Scores of TW | | Scores of JS | |
|---|---|---|---|---|
| | **Scores** | **T/F value** | **Scores** | **T/F value** |
| Gender | | 0.762 | | 1.825** |
| Male | 3.13±0.70 | | 3.91±0.80 | |
| Female | 3.18±0.64 | | 4.07±0.67 | |
| Age | | 0.126 | | 0.066 |
| <30 years | 3.16±0.74 | | 3.04±0.74 | |
| 30–45 years | 3.19±0.59 | | 3.06±0.67 | |
| >45 years | 3.16±0.63 | | 3.05±0.68 | |
| Years of working | | 0.267* | | 1.873* |
| <10 years | 3.15±0.69 | | 3.05±0.72 | |
| 10–20 years | 3.20±0.61 | | 3.03±0.68 | |
| >20 years | 3.17±0.67 | | 3.06±0.66 | |
| Educational level | | 0.396* | | 0.785** |
| Diploma | 3.16±0.54 | | 3.08±0.57 | |
| Bachelor's degree | 3.12±0.65 | | 3.02±0.68 | |
| Master's degree | 3.05±0.61 | | 3.96±0.46 | |
| Professional title | | 0.311 | | 0.655 |
| Junior staff | 3.05±0.72 | | 3.05±0.72 | |
| Middle staff | 3.03±0.68 | | 3.03±0.68 | |
| Senior staff | 3.04±0.66 | | 3.06±0.66 | |

Professional title is a comprehensive reflection of the registered public health workers' work ability and work position in the related field.
*P<0.05, **P<0.01.
JS, job satisfaction; OC, organisational commitment; PHWs, public health workers; TW, thriving at work.

**Table 2** Correlations among thriving at work, organisational commitment and job satisfaction among primary PHWs

| Variables | Mean±SD | TW | OC | JS |
|-----------|---------|-----|-----|-----|
| TW | 3.17±0.65 | 1 | — | — |
| OC | 2.93±0.73 | 0.715** | 1 | — |
| JS | 3.05±0.69 | 0.804** | 0.848** | 1 |

**P<0.01.
JS, job satisfaction; OC, organisational commitment; PHWs, public health workers; TW, thriving at work.

the mediating variable, and job satisfaction as the dependent variable. AMOS V.26.0 software was used to conduct the validation analysis of the hypothesised model, and the results showed that each path coefficient was statistically significant (p<0.001), as shown in figure 1. The model fit statistics were: ($\chi^2/df$)=3.416, Goodness-of-Fit Index (GFI)=0.982, adjusted GFI=0.953, Normed Fit Index=0.993, Comparative Fit Index=0.995, Incremental Fit Index=0.995 and root-mean-square error of approximation=0.063 (95% CI 0.042 to 0.086). All the above fit indices are within the acceptable range and the model fits well, as shown in table 3.

The bootstrap method was used to test the significance of the mediating role of organisational commitment between thriving at work and job satisfaction. Repeated samples were taken 2000 times, and 95% CIs were calculated. The results showed that the standardised effect value of the mediating effect of organisational commitment between thriving at work and job satisfaction was 0.422, 95% CI (0.276 to 0.561), and the standardised effect value of the total effect was 0.867, 95% CI (0.781 to 0.920), neither of which contained 0, indicating that the differences in all effect paths were statistically significant. The calculated mediated effect size was 48.7% of the total effect size (ie, the indirect effect value as a percentage of the total effect value), and the specific path coefficients of the model are shown in table 4.

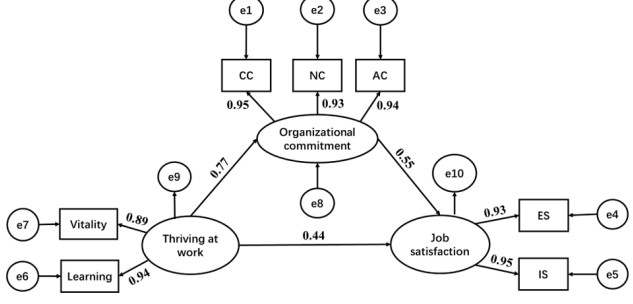

**Figure 1** Structural equation model of primary PHWs' thriving at work, organisational commitment and job satisfaction. AC, affective commitment; CC, continuance commitment; ES, external satisfaction; IS, internal satisfaction; NC, normative commitment; PHWs, public health workers.

# DISCUSSION

Based on the JD-R model, this study found that during the COVID-19 pandemic, primary PHWs in China had a moderate level of job satisfaction. Primary PHWs with adequate thriving at work are more likely to exhibit higher job satisfaction. In addition, thriving at work also can indirectly influence PHWs' job satisfaction through organisational commitment.

## The current status of thriving at work and job satisfaction among primary PHWs

Job satisfaction is an indispensable indicator of the quality of work performed by primary PHWs and has an important impact on their physical and mental health development.[41] However, during the pandemic in China, when small-scale outbreaks occurred in some areas, primary PHWs were on the front line of the fight against the epidemic and were required to respond quickly to their areas of responsibility.[42] Consequently, they were exposed to greater workload, physical injuries and mental stress caused by higher work demands, with insufficient work resources, which seriously deteriorated their job satisfaction.[43] The results showed that the thriving at work score of primary PHWs was 3.17±0.65. In addition, the score of job satisfaction was 3.05±0.69, which was lower than the scores of PHWs in Nigeria and Iran.[44 45] Meanwhile, the results of the ANOVA showed that there were statistically significant differences in the job satisfaction scores of primary PHWs in terms of gender, years of working and educational background. That is, primary PHWs who are male, have higher education, or have fewer years of working experience were at a relatively lower level of job satisfaction. The potential explanations could be the fact that grassroots positions offer fewer promotion and self-development opportunities, inadequate salaries and insufficient supportive measures, resulting in primary PHWs with master's degrees or higher being unable to give full play to their knowledge in healthcare practice, thus limiting their long-term career development and causing lower job satisfaction.[46] In addition, when employees have fewer years of working experience, their coping ability is still in the ascending stage, it is not possible to be very proficient in work operations and have limited access to work resources, especially in the heavy epidemic period, which to a certain extent reduces their job satisfaction.[47]

## Correlations and SEM among organisational commitment, thriving at work and job satisfaction among primary PHWs

Previous studies have shown a positive relationship between organisational commitment and job satisfaction among various industries, while such a relationship has not been confirmed among primary PHWs.[48] First, the results of Pearson analysis showed that thriving at work has a significantly positive correlation with job satisfaction ($\beta$=0.804, p<0.01). The integrated model of personal growth of 'thriving at work' suggests that a key output of thriving at work is the indicator of job satisfaction.[49] A meta-analysis that included 21 739 subjects found that teachers with a high level of thriving at work exhibited greater job satisfaction.[20] In addition, thriving at work is

**Table 3** Index evaluation system and fitting results of structural equation model adaptability

| Fit index | $\chi^2/df$ | GFI | AGFI | RMSEA | IFI | CFI | NFI |
|---|---|---|---|---|---|---|---|
| Theoretical model | <3 | >0.9 | >0.9 | <0.08 | >0.9 | >0.9 | >0.9 |
| Regression criterion | 3.416 | 0.982 | 0.953 | 0.063 | 0.995 | 0.995 | 0.993 |

AGFI, adjusted GFI; CFI, Comparative Fit Index; GFI, Goodness-of-Fit Index; IFI, Incremental Fit Index; NFI, Normed Fit Index; RMSEA, root-mean-square error of approximation.

one of the significant factors of work resources for healthcare occupations and health-related organisations. An individual's thriving focuses on two domains, vitality and learning, in daily work.[50] Primary PHWs can enhance learning through various types of training, to create a positive work atmosphere that is highly beneficial to both individuals and healthcare organisations. Moreover, vitality can help individuals adapt creatively to the difficulties of some situations, especially during the risky and challenging COVID-19 pandemic.[51] Second, organisational commitment maintained a highly positive relationship with job satisfaction ($\beta$=0.848, p<0.01). PHWs with high organisational commitment have a deep attachment to their healthcare units and a strong sense of responsibility and mission for their jobs. The correlations among the three components of organisational commitment and job satisfaction can be reflected as follows: if the primary PHWs exhibit higher affective commitment, they may have greater identification with the organisation's goals, values and culture, and higher satisfaction with their job.[52] Additionally, continuance commitment facilitates primary PHWs' loyalty to the organisation, which in turn enhances their satisfaction. Normative commitment ensures that the primary PHWs believe that they have a great responsibility to the organisation, and the greater the psychological satisfaction that active work can bring to the individual, the higher their job satisfaction.[53] Finally, there is a positive relationship between thriving and organisational commitment, and thriving on the content, atmosphere, and nature of their work promotes employees' attachment to the organisation, they are more likely to derive higher satisfaction from their work.[54]

The results of the SEM in this study also confirmed that organisational commitment partially mediated the relationship between thriving at work and job satisfaction, with a mediating effect size of −0.35, accounting for 57.38% of the total effect, indicating that organisational commitment serves as a significant factor affecting primary PHWs' level of job satisfaction. This is consistent with the result of Malgorzata's study, but his study participants were cross-cultural workers.[55] Therefore, the new relationships among thriving at work, organisational commitment and job satisfaction confirmed in this study offer the first validation in a group of Chinese healthcare providers. Furthermore, Bakker's study supports that organisational commitment is the result of a combination of many different job demands and job resources. Positive organisational commitment can have an impact on job satisfaction.[56] According to psychological contract theory, the cognitive structure of individuals determines how PHWs view their relationship with the organisation. This is considered to be their belief system, the obligations that exist between them and their work healthcare organisation.[57] Therefore, the mechanisms underlying the relationship indicate that thriving at work is a state of need and motivation that is critical to the internal psyche of the individual and that is easily activated and motivated by situational factors. A theoretical model of thriving at work developed by Spreitzer et al[58] suggests that when employees feel energised and full of learning in the workplace, they are energised and willing to invest more time, energy, and effort in their work, resulting in a high level of thriving at work. This will further stimulate their identification with their organisations (ie, affective commitment), which will

**Table 4** Pathway coefficients of structural equation model (n=601)

| Pathways | SE | 95% CI | SE | P value |
|---|---|---|---|---|
| Total effect | | | | |
| | 0.867 | 0.781 to 0.920 | | 0.002** |
| Direct effect | | | | |
| TW→JS | 0.445 | 0.269 to 0.640 | 0.098 | 0.001** |
| TW→OC | 0.765 | 0.662 to 0.844 | 0.046 | <0.01** |
| OC→JS | 0.552 | 0.364 to 0.726 | 0.096 | 0.001** |
| Indirect effect | | | | |
| TW→OC→JS | 0.422 | 0.276 to 0.561 | 0.070 | 0.001** |

**P<0.01.
JS, job satisfaction; OC, organisational commitment; TW, thriving at work.

lead to a higher level of centripetal force towards their careers and the groups they work with (ie, continuance commitment), and a greater willingness to engage in organisational activities and construction out of their own strong sense of responsibility and mission (ie, normative commitment).[59] Additionally, employees' commitment to the organisation inevitably affects employees' attitudes or feelings toward work, which in turn affects their job satisfaction. Positive work can bring psychological satisfaction to the individual, so the higher the employee's job satisfaction is, the higher the satisfaction with the job content, the nature of the job, and the atmosphere of the job.

This study explored the intrinsic association between thriving at work and job satisfaction by constructing an SEM with organisational commitment as a mediating variable. Thriving has been proven to promote job sustainability among primary PHWs and can have a direct impact on their job satisfaction, as well as an indirect impact on their job satisfaction by increasing the level of organisational commitment. This study provides a theoretical basis for better stabilising the talent pool of primary PHWs in China, improving the service quality of primary public health and medical care in China, and optimising the allocation of primary medical resources.

### Implications on policies or practices

The suggestions on the policies and practices can be made from the perspective of health policy-makers, as well as from that of the management of leaders. First, the health policymakers are advised to refer to the way of magnet hospitals, although magnet hospitals were originally applied to nursing initiatives to ameliorate the environment in which nurses practise. Specifically, policy-makers need to focus on providing a platform for staff career development, as well as providing supportive continuing education. Establishing a scientific and reasonable salary system and creating a positive organisational culture are conducive to mobilising and motivating primary PHWs, thus improving their job satisfaction. Second, for the leadership of employers, leaders should fully explore the potential advantages of primary PHWs in thriving at work, establish a relationship of mutual trust with their subordinates, respond positively to their opinions and suggestions, and cultivate their sense of organisational identity. Consequently, primary PHWs are more willing to integrate the organisation's development goals into their personal values and ideals, thus generating a higher level of organisational commitment, which is an intrinsic motivation for their proactive efforts.

### Study limitations

There are some limitations that cannot be ignored in this study. First, participants may have had recall bias when completing the scales. Second, we did not use a weighting method by the size of the provinces in the selection of primary healthcare units, which could be a potential source of sample selection bias. Third, physicians and nurses were the main participants in the fight against COVID-19 in other countries. This study has some implications for other countries, but the limitation in the generalisation of the results needs to be noted because of the differences in the target groups. Finally, the relationships among variables found in this cross-sectional study do not necessarily reflect causality. Future longitudinal studies with larger samples and incorporating more variables are needed to validate the results of this study.

## CONCLUSION

This study reveals that Chinese primary PHWs face high job demands and limited job resources, leading to their poor thriving and job satisfaction. Meanwhile SEM results found that thriving at work has a direct contribution to job satisfaction, while organisational commitment as a mediating variable also can increase job satisfaction. Therefore, in the context of the COVID-19 pandemic, health policy-makers should pay high attention to this group of primary PHWs and enhance their thriving and organisational commitment by regulating their job demands and providing sufficient job resources to effectively activate their job satisfaction and fulfilment, which is essential to maintain the job stability of primary PHWs.

**Author affiliations**
[1]School of Nursing and Rehabilitation, Cheeloo College of Medicine, Shandong University, Jinan, Shandong, China
[2]Department of Nursing, Peking Union Medical College Hospital Eastern Branch, Beijing, China
[3]Centre for Health Management and Policy Research, Cheeloo College of Medicine, Shandong University, Jinan, Shandong, China
[4]School of Public Health, Cheeloo College of Medicine, Shandong University, Jinan, Shandong, China

**Acknowledgements** We are particularly grateful to the coordinators who assisted us, especially all the primary PHWs who participated in this study and supported the study during the busy period of pandemic.

**Contributors** Conceived and designed the research: MY, DJ and OC. Performed the research: YJ, JW, LG and ZZ. Analysed the data: MY, YJ, MY and DJ; Contributed reagents/materials/analysis tools: YJ and BZ. Wrote the manuscript: MY. MY acts as guarantor for the work. All authors critically commented and approved the manuscript.

**Funding** This study was supported by The Natural Science Foundation of China (82172543); The Natural Science Foundation of Shandong Province (ZR2020MH006); The Key Research and Development Program of Shandong Province (2019GSF108198).

**Competing interests** None declared.

**Patient and public involvement** Patients and/or the public were not involved in the design, or conduct, or reporting, or dissemination plans of this research.

**Patient consent for publication** Not applicable.

**Ethics approval** This study was approved by Ethics committee(s) and IRB name: The ethic board of school of nursing and rehabilitation, Shandong university. Participants gave informed consent to participate in the study before taking part.

**Provenance and peer review** Not commissioned; externally peer reviewed.

**Data availability statement** Data are available on reasonable request.

ORCID iDs
Mo Yi http://orcid.org/0000-0003-4996-5797
Yuanmin Jia http://orcid.org/0000-0003-2787-0156

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
