## [Reviewer comments · BMJ Open]

ARTICLE DETAILS

TITLE (PROVISIONAL)	Relationships among thriving at work, organizational commitment and job satisfaction among Chinese frontline primary public health workers during COVID-19 pandemic: a structural equation model analysis
AUTHORS	Yi, Mo; Jiang, Di; Wang, Jingjing; Zhang, Zeyi; Jia, Yuanmin; Zhao, Baosheng; Guo, Lei; Chen, Ou

VERSION 1 – REVIEW

REVIEWER	Donald Pathman University of North Carolina at Chapel Hill School of Medicine, Cecil G. Sheps Center for Health Services Research
REVIEW RETURNED	09-Dec-2021

GENERAL COMMENTS	General comments This study draws together several important and interesting concepts about health care workers' experiences/responses to their jobs. The attempt to link the concepts of thriving at work, organizational commitment and job satisfaction is new, I believe. The survey measures used seem reasonable. I am not knowledgeable enough about structural equation modeling to assess this study's statistical approaches and interpretation of the findings. Further, there are many grammatical errors, rough sentences and wrong word choices throughout the manuscript. Unfortunately, the weaknesses of this manuscript's English writing make it difficult (impossible for me) to understand the authors' statements in the Introduction about the current state of the literature on clinicians' job satisfaction, self-fulfillment, and related concepts. The reader thus cannot understand and judge how this study contributes to the literature. Specific comments Page 4, lines 57-59. The COVID pandemic has caused millions of deaths worldwide, not tens of thousands. Page 5, lines 27-33. I agree with the many various statements here about the consequences of the pandemic for community-based clinicians. But with so many claims of important issues, it seems necessary to provide more citations. Page 5, lines 35-54. The text of this key paragraph is too brief, which leaves me as a reader unclear on the authors' conceptualization of job satisfaction, its measurement, relationship to self-fulfillment and relationships to hardships faced at work.
---

	Page 5, line 56. Should “behaviors” read instead “activities”? Page 6, line 4. Should “optimistic” read instead “positive”? Page 6, lines 23-25. Please provide citations for the claims that “highly close correlation between job satisfaction and thriving at work, both of which will be directly related to the level of quality of healthcare services.” Page 6, lines 30-31. Please explain what is meant by “as their ‘unilateral commitment’ to the organization increases.” Page 6, lines 31-35. If it is important for the reader to know that organizational commitment has the three components— affective, continuance and normative commitment—please explain what these three terms mean in the context of organizational commitment. I have no idea without an explanation. Page 6, lines 35-37. Please explain the relationship between intrinsic value and thriving at work. I see them as related but not identical concepts as this sentence seems to claim. Page 6, sentence beginning “Based on the original research findings . . .” Citations are needed here. Also, more explanation is needed because this is the first the reader encounters the concepts of “career stability” and “professional identity.” Page 7, lines 15-17. I do not understand the phrase “under the normalization of epidemic prevention and control.” Page 7. Section on “participants of the study.” What sampling frame was used, i.e., what list did you use of all workers in these 20 primary care units. As China is such a very large country, I imagine that it has thousands of primary care units and far more than 601 primary PHWs. Please explain how these 20 primary care units were identified and how representative they are of all units in the country. Also, please explain why “licensed doctor or nurses practitioner with qualification of public health” were excluded from this study. I would have thought that their experiences at work during the pandemic would be very relevant to this study given the study’s stated aims. Further, the 92% response rate is far higher than I have ever seen in a voluntary survey of workers in any field. Was this study truly voluntary? Page 7, lines 9-11. In the list of references, reference #21, please provide a citation from Weiss, Dawis England and Lofquist, not a citation from other authors who cited Weiss et al. Further, I assume that a Chinese version of their instrument was used in this study, so please provide a citation to that translation and its validation. Please also report if this job satisfaction instrument has been used successfully with healthcare workers. Page 8, line 19. Unless the editor requests otherwise, the Cronbach’s alpha from this study’s data should be presented in the Results section of the paper. Same issue for the alphas reported on line 41 and lines 58-60. Page 8, lines 33-35. For the sentence, “The Cronbach alpha for each dimension of the scale was 0.87, 0.75, and 0.79”, please clarify
--	---

	if these are alphas reported in the literature (and if so, provide citations) or if these are alphas derived in this study's data. If the latter, this information belongs in the Results section. Page 9, lines 25-26. Please provide the name of the university of the Ethics Committee, and provide the study approval number and date. Page 9, "data collection." Please state if this was a written survey. Also, I do not see that confidentiality or anonymity was promised to the subjects and that they were assured that their responses would not be reported to their employers. Was confidentiality promised? Page 10, line 18-20. The statement is "A total of 601 primary PHWs . . . were surveyed." The Methods states that 650 PHWs were surveyed but that 601 responded. Page 10, second on "general demographic." Please provide information on the disciplines of study participants. Since some or all physicians and nurse practitioners were excluded, and the discussion indicates that subjects were responsible for running residential nucleic acid testing and disinfecting the environment, I am not clear on the types of health care workers that were studied. Page 10 line 32 through Page 11 line 10 and Table 1. Why are bivariate associations for organizational commitment not presented in the text or in Table 1? I would expect to see this information together with the bivariate analyses presented for thriving a work and job satisfaction and before the findings of the subsequent correlation analyses. Page 14, lines 34-36. The statement "Adequate thriving at work can directly enhance job satisfaction." strongly implies causality in this study's findings from a cross-sectional study. Similarly, causation is too strongly implied on page 17, lines 4-6 in the sentence, "This study also found that the relationship between thriving at work and job satisfaction was also influenced by organizational commitment to some extent." See "Boelen KA, Pearl J. Eight myths about causality and structural equation models" in Handbook of Causal Analysis for Social Research. Chapter 15, 301-328. Springer. 2013. The Limitations appropriately cautions the reader that this is a cross-sectional study which limits claims of causation. Page 14, line 44. I question the validity of the statement "Job satisfaction is an important indicator of the quality of work done by primary PHWs." It is clear from prior studies that job satisfaction correlates with quality of work, but the association is not strong enough to claim that it is an important indicator of quality of work.
--	--

REVIEWER	Walter Sermeus Catholic University Leuven, Centre for Health Services and Nursing Research
REVIEW RETURNED	17-Dec-2021

GENERAL COMMENTS	It is an interesting study, well formulated and performed. The main improvement for the manuscript can be done in the literature review and referring to the state of the literature. I see no reference to the work of Bakker and Schaufeli related to work engagement what seems to me a similar concept as thriving at work. Work engagement is widely used to express the positive behavior in
--

	work. The Job satisfaction literature is often referring to the work of Frederic Herzberg in the '50ties on motivational factors (the positive factors on work behavior) and the 'hygiene factors' related to the negative feelings of job dissatisfaction and burnout. On the relationships between work environment and job outcomes, the job demand support model (Karacek) or the Job demand resources model (Bakker, Schaufeli) are often used. So it would be good to relate the used concepts and models in this manuscript to the larger domain of similar and frequently used concepts and models. The findings of the study are coherent with earlier findings in applying these models, so the reference might also enhance the discussion section as well. p7 Aim and objectives of the study are clear and well formulated p7 It is unclear which professionals were included in the study. We find in the results that they are mainly female and college based. But what we want to know is the distribution among professions (nurses, physicians, allied health professionals, ...) p9 r8 it is unclear what is meant by 'an independent and anonymous self-administered manner': please be more concrete and specific p10 r59 'the group of lower professional titles... " unclear p11 table 1 Analysis should be explained or repeated. I checked the first analysis (Gender and TW). I learn from the descriptive that you have 82,9% female (means 498 F and 103 M). We have average scores 3.13 and 3.18 and SD (0.70 and 0.64). When I do a unpaired T-test I receive a t-test of 0,71 (df 599) and a p-value of 0,478 which is not significant. There might be small differences in the data used, but this score cannot be significant: check again. For JS, I find a t-statistic of 11.18 and a significant result (<0,0001) Please check all analyses and results p23 the figure on the SEM is not clear. It should summarize the information in the model of table 4 on p14 (with arrows + estimates on how the effect is explained)
--	--

REVIEWER	Beibei Yuan Peking University, China Center for Health Development Stud
REVIEW RETURNED	21-Dec-2021

GENERAL COMMENTS	 1. The topic on the relationship among these concepts have limited implications on policies or practices. 2. For some key arguments, the references are lacking, like in line 23-25, the evidences on the relationship between thriving at work and healthcare quality. 3. The hypothesis on relationship among three concepts is not clearly presented in background to justify the motivation of this study. 4. The sampling process is not detailed. Which provinces are covered? The characteristics of covered areas? How many insitutions are covered? All these determines the applicability of findings of this study. 5. In discussion, the construct of equaton model, the theory basis and the comparison with other studies are all relatively weak. 6. No clear and relevant policy implications can be got from these findings.
--

VERSION 1 – AUTHOR RESPONSE

Reviewer: 1

Dr. Donald Pathman, University of North Carolina at Chapel Hill School of Medicine

Comments to the Author:

Specific comments

Page 4, lines 57-59. The COVID pandemic has caused millions of deaths worldwide, not tens of thousands.

Response: Thank you for your suggestion. After reviewing the latest data of death of COVID-19 pandemic, we have revised it to the “millions of deaths”.

Page 5, lines 27-33. I agree with the many various statements here about the consequences of the pandemic for community-based clinicians. But with so many claims of important issues, it seems necessary to provide more citations.

Response: Thanks to your suggestion, we have provided more relevant citations in the paragraph of ‘Introduction’ part in revised manuscript to complement the effects of the epidemic on the physiological and psychological aspects for the primary PHWs, as well as highlight the importance of the need to focus on the physical and mental condition of this group¹.

The primary PHWs struggling to fight the epidemic are overloaded with work intensity and great risk of occupational exposure, and the epidemic has caused a significant burden on the primary PHWs, affecting their psychological health, specifically through the existence of psychological problems such as fear, suspicion, and neurosis, and undergoing great pressure and tough tests^{2, 3}.

Page 5, lines 35-54. The text of this key paragraph is too brief, which leaves me as a reader unclear on the authors’ conceptualization of job satisfaction, its measurement, relationship to self-fulfillment and relationships to hardships faced at work.

Response: The conceptualization of job satisfaction is “Job satisfaction is the subjective psychological state of employees’ physiological and psychological perceptions of the work environment and the work itself^{4, 5}.”

Employees with higher job satisfaction are mostly enthusiastic about their daily work content and are able to feel more positive psychology⁶. Even in the face of hardship times like epidemics, they are able to cope well, thus showing a higher level of self-fulfillment, this is the suitable description of the relationship to self-fulfillment and relationships to hardships. The measurement named “*Minnesota Satisfaction Questionnaire*” to exam an employee’s job satisfaction is divided into two parts: internal

(personal situation) and external (working environment), and the quantitative data are used to measure the job satisfaction of the study participants⁷.

Page 5, line 56. Should “behaviors” read instead “activities”?

Response: Thank you for your suggestion, we have changed the word “behavior” to “activities”.

Page 6, line 4. Should “optimistic” read instead “positive”?

Response: Thank you for your suggestion, we have changed the word “optimistic” to “positive”.

Page 6, lines 23-25. Please provide citations for the claims that “highly close correlation between job satisfaction and thriving at work, both of which will be directly related to the level of quality of healthcare services.”

Response: Thank you for your suggestions. There is a high positive correlation between job satisfaction and thriving at work, and that both have some influence on the quality of healthcare services. Zhou’s and Huo’s studies^{8, 9} have verified the correlation between these two. We have added relevant citations to the manuscript.

Page 6, lines 30-31. Please explain what is meant by “as their ‘unilateral commitment’ to the organization increases.”

Response: The concept of organizational commitment was first introduced by Becker (1960). He defined commitment as the tendency to maintain "consistency of activity" resulting from unilateral input (side-bet)¹⁰. In an organization, such ‘unilateral commitment’ (side-bet) can refer to anything of value, such as benefits, energy, skills already acquired that can only be used in a particular organization, etc¹¹. He sees organizational commitment as a psychological phenomenon that encourages employees to stay with the organization and participate wholeheartedly in its work as their "side-bet" to the organization increases.

Page 6, lines 31-35. If it is important for the reader to know that organizational commitment has the three components— affective, continuance and normative commitment—please explain what these three terms mean in the context of organizational commitment. I have no idea without an explanation.

Response: According to the model of organizational commitment released by John P. Meyer and Natalie J. Allen^{12, 13}, the organizational commitment could be divided into three components: affective, continuance and normative commitment. Affective commitment expressed as identification with and acceptance of organizational goals and values, which positively influences job performance. Continuance commitment manifests itself as an individual's willingness to stay with the organization

as their commitment to the organization increases and they become aware of the sunk costs. Normative commitment manifests itself as employees agreeing with the general ethical standards for treating their work and feeling a responsibility to stay with the organization. Due to the limitation of word count of the introduction, we cannot explain each subcomponent one by one. If necessary and if the editor agrees, we will add an explanation of these three dimensions of organizational commitment in the manuscript.

Page 6, lines 35-37. Please explain the relationship between intrinsic value and thriving at work. I see them as related but not identical concepts as this sentence seems to claim.

Response: Buchanan and Porter raised a new issue initially in 20 Century that organizational commitment is one of the important components of employees' work attitudes, which is an individual's identification with the goals and values of the organizations and is influenced by the intrinsic values of the individual subject as well as by external motivational mechanisms^{14, 15}. The intrinsic values mentioned in this section of manuscript are the beliefs and preferences that employees hold about the work they do. Porth believes that human thriving can be seen as a measure of perceived progress and growth, effectively helping individuals to develop in a positive direction. Spreitzer and colleagues also developed a theoretical model of thriving at work, which explains how certain individual characteristics (e.g., intrinsic perceived value, knowledge, positive affect), interpersonal/relational characteristics (e.g., support, trust), contextual features (e.g., job autonomy, climate of trust), and agentic work behaviors (e.g., task focus, exploration) lead to thriving at work¹⁶. Thus, Abid concluded from a survey of 936 employees that the improvement of employees' intrinsic value contributes to the improvement of their own organizational commitment, which indirectly motivates them to become motivated and energetic people¹⁷. This is a concrete manifestation of thriving, i.e., learning and vitality.

Therefore, I wrote in the original manuscript that 'The level of organizational commitment increases with intrinsic value, which is another expression of thriving at work' may need to be more specifically described.

Page 6, sentence beginning "Based on the original research findings . . ." Citations are needed here. Also, more explanation is needed because this is the first the reader encounters the concepts of "career stability" and "professional identity."

Response: Thank you for your suggestion, we have added relevant citations and explained "career stability" and "professional self-identity" correspondingly. According to the relevant literature, "Career stability" means that the individual has a stable job for a certain period of time and does not have the idea of leaving easily¹⁸. "Professional self-identity" refers to the individual's perception of the goals

and values of the occupation he or she is engaged in, and the consistency with society's evaluation and expectations of the occupation^{19, 20}. Meanwhile, we have added an explanation of the terms' definitions of "career stability" and "professional identity." to this section of the revised manuscript.

Page 7, lines 15-17. I do not understand the phrase “under the normalization of epidemic prevention and control.”

Response: Thank you for your question. The phrase “under the normalization of epidemic prevention and control” means that the COVID-19 pandemic in China is currently at a normal stage with small scale spread in particular area. We have modified the phrase into “during the stabilization period of the COVID-19 epidemic” to make it more precise so that readers can understand the meaning easily.

Page 7. Section on “participants of the study.” What sampling frame was used, i.e., what list did you use of all workers in these 20 primary care units. As China is such a very large country, I imagine that it has thousands of primary care units and far more than 601 primary PHWs. Please explain how these 20 primary care units were identified and how representative they are of all units in the country. Also, please explain why “licensed doctors or nurse practitioners with qualification of public health” were excluded from this study. I would have thought that their experiences at work during the pandemic would be very relevant to this study given the study's stated aims.

Further, the 92% response rate is far higher than I have ever seen in a voluntary survey of workers in any field. Was this study truly voluntary?

Response: This study used a convenience sampling method to select the targeting participants (registered public health workers) in primary care units in the northern provinces of China, spread across Shandong, Hebei, and Henan and Shanxi provinces in China. The method of determining the units was that we randomly selected 20 from the list with hundreds of units recommended by every provincial health care commission for the survey, after obtaining their written consent. Thus, our primary care units were identified.

What's more, the reason why we maintain a high response rate may be due to the fact that this study focused on the public health worker population, and was able to express the specific demands of this group during the pandemic, as well as aimed to contribute to the resolution of the difficulties that existed in the daily work of primary PHWs. In addition, the strong support from the public health workers' organizations/units, which enhanced the pre-study outreach, was a key factor in maintaining a high response rate among the participants in this study.

It is worth mentioning that after 44 years of reform and opening up in China, the economic growth rate in northern China is much lower than that in the southern part, as well as the state financial investment and financial support is lower than that in the southern region, resulting in the primary care resources and health service capacity in this region is also much weaker than that in the south. Therefore, these 20 primary care units are representative of the less developed regions of China.

Besides, this study focuses on the current status of thriving at work, job satisfaction, and organizational commitment of primary PHWs in northern provinces of China and their intrinsic relationships, which can help to better understand the primary public health system in less developed regions of China and provide health policymakers with further emphasis on strengthening the primary public health workforce, increasing the introduction of experts and scholars, and increasing the funding on public health, as well as designating other relevant policies.

Page 7, lines 9-11. In the list of references, reference #21, please provide a citation from Weiss, Dawis England and Lofquist, not a citation from other authors who cited Weiss et al. Further, I assume that a Chinese version of their instrument was used in this study, so please provide a citation to that translation and its validation. Please also report if this job satisfaction instrument has been used successfully with healthcare workers.

Response: Thank you for your suggestion, we have noted this unnecessary mistake in the manuscript. The original reference to the Minnesota Satisfaction Questionnaire (MSQ) in the manuscript has been changed to the reference of the scale designer "Weiss, Dawis England and Lofquist"²¹, instead of other authors. Besides, we also conducted a database search for the Chinese version of the MSQ and found literature reports of Chinese versions of the MSQ and some studies of its application in healthcare workers, like the group of nurses or physician²²⁻²⁴.

Page 8, line 19. Unless the editor requests otherwise, the Cronbach's alpha from this study's data should be presented in the Results section of the paper. Same issue for the alphas reported on line 41 and lines 58-60.

Response: Thanks to your suggestion, we have adjusted the presentation of the Cronbach's alpha value for the three scales (Thriving at work, job satisfaction and organizational commitment) from the **Methods section** of original manuscript to the **Results section** of the revised manuscript: Cronbach's alpha value for the job satisfaction scale is 0.919, the Cronbach's alpha value for the organizational commitment scale is 0.917, and the Cronbach's alpha value for the euphoria scale is 0.896.

Page 8, lines 33-35. For the sentence, "The Cronbach alpha for each dimension of the scale was 0.87, 0.75, and 0.79", please clarify if these are alphas reported in the literature (and if so, provide citations) or if these are alphas derived in this study's data. If the latter, this information belongs in the Results section.

Response: Thank you for your suggestions, I understand what you mean. The Cronbach alpha value presented in the **Method** section of manuscript is based on the results of the previous study and

indicates that the high reliability of the scales is acceptable and mature, they are not the Cronbach alpha value of the three scales in this study. The measurement instruments were selected by comparing the use of different instruments among employees (including reliability, stability, and their impact), and the different measurement (scales) corresponding to each of the three variables (job satisfaction, organizational commitment, thriving at work) were determined only after group discussion. The Cronbach's alpha value of the three different dimensions of the Organizational Commitment Scale presented in this study were actually derived from a Chinese paper²⁵. We selected a high-quality literature that reported the Cronbach's alpha value for three different components of the Organizational Commitment Scale²⁶: the overall Cronbach's alpha of each component is: 0.928 (affective commitment), 0.802 (continuance commitment), 0.859 (normative commitment). We have provided it as the citation in the revised manuscript.

Page 9, lines 25-26. Please provide the name of the university of the Ethics Committee, and provide the study approval number and date.

Response: Thanks to your suggestion, we have added the name of the university of ethics committee, as well as the approval number and the date of obtaining the number in the "data collection" section, the details can be seen in the revised manuscript.

Page 9, "data collection." Please state if this was a written survey. Also, I do not see that confidentiality or anonymity was promised to the subjects and that they were assured that their responses would not be reported to their employers. Was confidentiality promised?

Response: In this study, after obtaining informed consent from the participants, the participants were asked to fill out the paper questionnaire independently. After completing the questionnaire, our team members checked whether there were any omissions or errors that would lead to invalidation of the questionnaire, and the paper questionnaire was kept by our staff after verification. Therefore, this study is a written survey. In addition, our team promises that the questionnaires filled in do not contain personal privacy information, and are anonymous, and do not show the names of participants, workplace names, etc. Furthermore, the data obtained will not be given to their employers to ensure that the data obtained reflects the true thoughts of the participants to the greatest extent possible.

Page 10, line 18-20. The statement is "A total of 601 primary PHWs . . . were surveyed." The Methods states that 650 PHWs were surveyed but that 601 responded.

Response: Thanks to your suggestion, our team invited a total of 650 participants who met the inclusion and exclusion criteria, and eventually received responses from 601 participants who agreed

and completed the relevant scales. Therefore, we finally used the data obtained from the 601 participants for the follow-up analysis. Also, we modified the data in the statement in the manuscript to keep it consistent throughout the text, like “A total of 601 primary PHWs... were surveyed” you mentioned.

Page 10, second on “general demographic.” Please provide information on the disciplines of study participants. Since some or all physicians and nurse practitioners were excluded, and the discussion indicates that subjects were responsible for running residential nucleic acid testing and disinfecting the environment, I am not clear on the types of health care workers that were studied.

Response: Only the registered public health workers were investigated in this study. In China, public health workers are another group of healthcare provider, different with the other professional group, like doctors, pharmacists and nurses. They are mainly responsible for epidemiological investigation, assisting personnel home-isolation, community decontamination, and nucleic acid collection in the community or other primary organization during the pandemic of COVID-19²⁷. While physicians are primarily responsible for patient treatment and rehabilitation, nurses are responsible for providing physical and psychological care services to patients. The data from national government website indicate that the number of public health workers in China is small than other group pf healthcare provider, therefore the social and research attention to them prior to the pandemic was far low, and much lower than that of the physician and nurse populations²⁸. A recent study found that the prevalence of probable depression and anxiety among public health workers during the pandemic in China was 21.3% and 19.0%, respectively²⁹. Therefore, this study focused only on the group of public health workers, so as to better improve the public health service system, reduce the occurrence of infectious diseases, and provide comprehensive and life-cycle health management for the residents in remote rural areas.

Page 10 line 32 through Page 11 line 10 and Table 1. Why are bivariate associations for organizational commitment not presented in the text or in Table 1? I would expect to see this information together with the bivariate analyses presented for thriving a work and job satisfaction and before the findings of the subsequent correlation analyses.

Response: Because Table 1 is a one-way ANOVA on the main variables, organizational commitment is a mediating variable (a non-significant and secondary variable) in this study, and because of the word limit of the manuscript, we only analyzed the demographic factors of variable X (thriving at work) and variable Y (job satisfaction) when conducting the one-way ANOVA. However, we did a bivariate association of the three variables in Table 2, i.e., we performed Pearson correlations for the three variables.

Page 14, lines 34-36. The statement “Adequate thriving at work can directly enhance job satisfaction.” strongly implies causality in this study’s findings from a cross-sectional study.

Similarly, causation is too strongly implied on page 17, lines 4-6 in the sentence, “This study also found that the relationship between thriving at work and job satisfaction was also influenced by organizational commitment to some extent.”

See “Boelen KA, Pearl J. Eight myths about causality and structural equation models” in *Handbook of Causal Analysis for Social Research*. Chapter 15, 301-328. Springer. 2013.

The Limitations appropriately cautions the reader that this is a cross-sectional study which limits claims of causation.

Response: Thank you for your suggestion. After reviewing the e-book you recommended, *Handbook of Causal Analysis for Social Research*, we realized that the original manuscript was not rigorous in writing the relationships between some variables. Because cross-sectional studies cannot strongly infer causal relationships between variables, we have revised some statements in the revised manuscript that exist to describe the relationships between variables accordingly, such as the revised sentence ‘primary PHWs with adequate thriving at work are more likely to exhibit higher job satisfaction’.

Page 14, line 44. I question the validity of the statement “Job satisfaction is an important indicator of the quality of work done by primary PHWs.” It is clear from prior studies that job satisfaction correlates with quality of work, but the association is not strong enough to claim that it is an important indicator of quality of work.

Response: Thank you for your suggestion. After reviewing some articles which focus on the correlation between quality of work and job satisfaction in the group of employees, we found that there is a high positive correlation between job satisfaction and job quality, and that when employees are satisfied with their job content, it definitely produces good job quality^{30, 31}. After the discussion within our research team, we concluded that there are many dimensions or indicators to measure the quality of employees' work, and job satisfaction is only one of the many indicators, and it is not enough to say "is an important indicator". Therefore, we changed the sentence into "Job satisfaction is one of the indicators to measure the quality of work of primary public health workers".

Reviewer: 2

Dr. Walter Sermeus, Catholic University Leuven

Comments to the Author:

It is an interesting study, well formulated and performed.

The main improvement for the manuscript can be done in the literature review and referring to the state of the literature. I see no reference to the work of Bakker and Schaufeli related to work engagement what seems to me a similar concept as thriving at work. Work engagement is widely used to express the positive behavior in work. The Job satisfaction literature is often referring to the work of Frederic Herzberg in the '50ties on motivational factors (the positive factors on work behavior) and the 'hygiene factors' related to the negative feelings of job dissatisfaction and burnout. On the relationships between work environment and job outcomes, the job demand support model (Karacek) or the Job demand resources model (Bakker, Schaufeli) are often used.

So it would be good to relate the used concepts and models in this manuscript to the larger domain of similar and frequently used concepts and models.

The findings of the study are coherent with earlier findings in applying these models, so the reference might also enhance the discussion section as well.

Response: Thank you for your suggestion. We note that thriving at work and work engagement are similar concepts. thriving at work is a positive psychological state characterized by a joint sense of vitality and learning, and work engagement is a positive, energetic, persistent, and pervasive affective-cognitive state characterized by energy, dedication, and focus. Both emphasize the positive psychological state that employees exhibit during the work process³². Thriving at work, on the other hand, not only emphasizes vitality, but also highlights the contribution of the learning component to the employee's thriving, which is more specific in describing positive psychology than the concept of work engagement. Employees who are thriving experience personal growth by feeling energized and alive (i.e., vitality) and by having a sense of continually acquiring and applying knowledge (i.e., learning)³³. This is the reason why this study selected the indicator of 'thriving at work', instead of 'work engagement'.

In addition, we refer to other literature, especially the literature published by Bakker and Karacek. Finally, after comparing the job demand-support model and the job demand-resource model, we selected the job demand-resource model as the theoretical model which was more compatible in this study, and added the description of that model in literature review part of the revised manuscript. We provided the relevant citation in the same time.

The job demands-resources (JD-R) model proposed by Bakker and his colleagues was used to support the theory of this study. A central assumption of the model in this study is that although each job may have its own specific characteristics related to job burnout and job satisfaction, these characteristics can be classified into two broad categories: namely, job demands and job resources. Bakker's proposed job demands-resources (JD-R) model was used to support the theory of this study³⁴. A central assumption of the model in this study is that although each occupation may have its own specific job characteristics related to job burnout and job satisfaction, these characteristics can be classified into two broad categories: namely, job demands and job resources. The JD-R model reveals two underlying psychological processes that lead to job satisfaction and burnout: excessive job demands keep individuals in a constant state of fatigue resulting in emotional exhaustion and a

continuous decrease in satisfaction. In contrast, lack of job resources leads to individual dehumanization and reduced sense of accomplishment.

Due to the requirement of word limit from the editor, we briefly introduce the model in the revised manuscript, as well as add the JD-R model with the concepts of this study to elaborate the rationale of using the model. We consider the increased workload and physical and mental exhaustion of primary PHWs from the pandemic as an important component of job demands, the organizational commitment and thriving as a reflection of job resources, which interact with each other and are reflected in the study outcome "job satisfaction". Therefore, thank you for the recommended model, which makes this study more substantial and logical.

p7 Aim and objectives of the study are clear and well formulated

Response: Thanks for your review.

p7 It is unclear which professionals were included in the study. We find in the results that they are mainly female and college based. But what we want to know is the distribution among professions (nurses, physicians, allied health professionals, ...)

Response: The majority of the study participants in this study were female, accounting for 82%. Additionally, most of their education background was college degree. This is the baseline characteristic of the participation in this study. Importantly, this study only included the group of primary PHWs and did not include physicians, nurses, or other allied health professionals. Unlike in Western countries, in China, public health workers are a separate group of healthcare providers from physicians and nurses, and can only work in the field of public health after obtaining a bachelor's degree, passing a graduation exam, and getting a national public health worker certification. The duties of public health workers include epidemiological investigation, vaccination, epidemic isolation of people, community disinfection and specimen collection. Therefore, the professions distribution is only the public health workers, instead of the nurses, physicians or other allied health professionals.

p9 r8 it is unclear what is meant by 'an independent and anonymous self-administered manner': please be more concrete and specific

Response: This study was conducted by sending an email to a random sample of 650 primary public health workers from a list of 20 primary units to solicit their willingness to participate our study. After obtaining their informed consent, we informed them face-to-face of the requirements for completion, and then the participants filled out the paper questionnaire independently. After completing the questionnaire, our team members checked it, and if there was no missing information, the questionnaire was administered by us for safekeeping. This is the concrete meaning of the word 'independent' and 'self-administered'. In addition, our team promised that the completed questionnaires are anonymous and do not show the name, workplace name, living address and other privacy information of participants. In addition, the data obtained will not be given to their employers to ensure that the data obtained reflects the true thoughts of the participants to the greatest extent possible. This is the specific description of the 'anonymous'

p10 r59 'the group of lower professional titles... " unclear “较低职称的人群” 不清楚

Response: Professional title refers to the professional level and position of registered public health workers, which is a comprehensive reflection of their work experience and work ability, and is recognized by the Ministry of Human Resources and Social Security of China. In China, this professional title system has 5 levels from junior to senior: level 1 and 2 are junior, level 3 is intermediate, and level 4 and 5 are senior titles. The "low professional title" in your question is that someone with level 1 or level 2 among primary public health workers. And we changed it into 'professional levels'

p11 table 1 Analysis should be explained or repeated. I checked the first analysis (Gender and TW). I learn from the descriptive that you have 82,9% female (means 498 F and 103 M). We have average scores 3.13 and 3.18 and SD (0.70 and 0.64). When I do a unpaired T-test I receive a t-test of 0,71 (df 599) and a p-value of 0,478 which is not significant.

There might be small differences in the data used, but this score cannot be significant: check again.

For JS, I find a t-statistic of 11.18 and a significant result (<0,0001)

Please check all analyses and results

Response: Thank you for your suggestion, we re-analyzed and checked all the analyses and results in this study and found two errors in the analysis of one-way ANOVA in Table 1, but no errors in the rest. Some of the errors may be due to different statistical software or different versions. Therefore, we have revised the relevant data in the revised manuscript, and we ask you to check them.

p23 the figure on the SEM is not clear. It should summarize the information in the model of table 4 on p14 (with arrows + estimates on how the effect is explained)

Response: Thanks to your suggestion, we have included a clear structural equation model (SEM) figure in the revised manuscript. The figure below shows the direction of the arrow between each latent variable in this study with its estimated value, which facilitates the reader to understand the path coefficients of the variables.

Abbreviations: CC, continuance commitment; NC, normative commitment; AC, affective commitment; IS, internal satisfaction; ES, external satisfaction.

Figure 1 Structural equation model of primary PHWs’ thriving at work, organizational commitment, job satisfaction

Reviewer: 3

Dr. Beibei Yuan, Peking University

Comments to the Author:

1. The topic on the relationship among these concepts have limited implications on policies or practices.

Response: Thanks to your suggestion, our findings suggest that the relationship between these three main concepts is that thriving at work can affect job satisfaction either directly or indirectly through organizational commitment, and all three relationships are highly positively related. As the link

between employees and the organization, organizational commitment plays an important mediating role between thriving at work and satisfaction. To address the relationship of these three concepts, we have provided their implications for policy or medical practice in the revised manuscript.

Mechanisms underlying the relationship: thriving at work is a state of need and motivation that is critical to the internal psyche of the individual and that is easily activated and motivated by situational factors such as the content of the work itself, employer leadership, or workplace rules³⁵. Spreitzer and colleagues developed a theoretical model of thriving at work³⁶, which suggests that when employees feel energized and full of learning in the workplace, they are energized and willing to invest more time, energy, and effort in their work, resulting in high level of thriving at work. This will further stimulate their identification with their organizations (i.e., affective commitment), which will lead to a higher level of centripetal force toward their careers and the groups they work with (i.e., continuance commitment), and a greater willingness to engage in organizational activities and construction out of their own strong sense of responsibility and mission (i.e., normative commitment). Recent studies by Kleine have confirmed a positive relationship between thriving at work and organizational commitment (RC=0.63)³⁷. Additionally, employees' commitment to the organization inevitably affects employees' attitudes or feelings toward work, which in turn affects their job satisfaction. Positive work can bring psychological satisfaction to the individual, so the higher the employee's job satisfaction, i.e., higher satisfaction with the job content, the nature of the job, and the atmosphere of the job. The relationship between the three concepts is presented above. Due to the word limit, I have to make a brief statement of the relationship between the three concepts in the manuscript.

Implications on policies or practices: Creating a healthy work environment is an effective way to increase job satisfaction, reduce burnout and turnover, and stabilize the primary PHWs. Suggestions can be made from the perspective of health policymakers, as well as from the perspective of leaders.

1. Firstly, the health policymakers are suggested that they can refer to the way of magnet hospitals. Although the magnet hospital originally was a new management approach applied to nursing for initiatives to ameliorate the environment in which nurses practice. Several studies have shown that the magnetic environment can effectively reduce nurses' turnover and thus improve the quality of care in hospitals³⁸⁻⁴⁰. Specifically, health policymakers should focus on providing a platform for staff development and training programs for staff competencies, as well as helping staff with career planning and providing supporting human resource development programs and training to encourage them to participate in continuing education⁴¹. Establishing a scientific and reasonable salary management system and creating a positive hospital culture is conducive to mobilizing and motivating employees, thus improving their job satisfaction⁴².

2. Secondly, for the leadership of employers, the leaders should fully explore the potential advantages of primary PHWs in thriving at work, establish a relationship of mutual trust with their subordinates,

respond positively to their opinions and suggestions, and cultivate their sense of organizational identity⁴³. As a result, PHWs are more willing to integrate the organization's development goals into their personal values and ideals, thus generating a higher level of organizational commitment, which is an intrinsic motivation for their spontaneous and proactive work.

2. For some key arguments, the references are lacking, like in line 23-25, the evidences on the relationship between thriving at work and healthcare quality.

Response: Thanks to your suggestion, we have provided the evidence and some statements in the revised manuscript to argue for the relationship between the thriving at work and healthcare quality. The relationship between them showed a highly positive association, the achievement of personal thriving by healthcare workers is important for improving the quality of healthcare services. Liu conducted a meta-analytic review based on the socially embedded model of thriving at work constructed by Spreitzer⁴⁴, which showed that Thriving at Work as a positive affective resource can stimulate personal work enthusiasm and satisfaction by promoting proactive work behaviors such as continuous learning and positive vitality. When the model is applied to the group of healthcare providers, and when individual behaviors evolve into group behaviors, it contributes to the improvement of the quality of healthcare in society as a whole.

3. The hypothesis on relationship among three concepts is not clearly presented in background to justify the motivation of this study.

Response: Thanks to your suggestion, we have added the study hypothesis about the three concepts in the last paragraph of the "**introduction**" section of the revised manuscript. Based on the arguments we mentioned in the Introduction part, we hypothesized that (1) thriving at work and organizational commitment are protective factors for job satisfaction among primary PHWs, and (2) thriving at work would indirectly and directly affect job satisfaction through organizational commitment.

4. The sampling process is not detailed. Which provinces are covered? The characteristics of covered areas? How many insitutions are covered? All these determines the applicablity of findings of this study.

Response: This study used a convenience sampling method to select the targeting participants (registered public health workers) in primary care units in the northern provinces of China, spread across Shandong, Hebei, and Henan and Shanxi provinces in China. The method of determining the

units was that we randomly selected 20 from the list with hundreds of units recommended by every provincial health care commission for the survey, after obtaining their written consent. Thus, our primary care units were identified.

It is worth mentioning that after 44 years of reform and opening up in China, the economic growth rate in northern China is much lower than that in the southern part, as well as the state financial investment and financial support is lower than that in the southern region, resulting in the primary care resources and health service capacity in this region is also much weaker than that in the south. Therefore, these 20 primary care units are representative of the less developed regions of China. Besides, this study focuses on the current status of thriving at work, job satisfaction, and organizational commitment of primary PHWs in northern provinces of China and their intrinsic relationships, which can help to better understand the primary public health system in less developed regions of China and provide health policymakers with further emphasis on strengthening the primary public health workforce, increasing the introduction of experts and scholars, and increasing the funding on public health, as well as designating other relevant policies.

5. In discussion, the construct of equation model, the theory basis and the comparison with other studies are all relatively weak.

Response: Thanks to your suggestion, we develop a deep review of the SEM results in the discussion part, and the SEM was constructed to give us more insight into the intrinsic relationship between the three variables. Also, we cited the (Job Demand-Resource, JD-R) model as the theoretical model in the introduction and discussion section, while using the model to support the results and findings of this study. The JD-R model reveals two potential psychological processes that lead to burnout and affect job satisfaction among primary PHWs: steeply increasing workload and excessive epidemic prevention requirements during pandemic cause individuals to be continuously fatigued and thus emotionally depleted, while lack of job resources leads to a decrease in individual dehumanization and fulfillment, which adversely affects job satisfaction. Therefore, this theoretical model provides a relevant reference for policymaking in this study.

In addition, we added a comparison with other studies in the discussion part to highlight the innovation of this study compared to other previous studies, as well as to focus on primary PHWs with the aim of improving their job satisfaction and reducing employee turnover, and to solidify the significance of this study.

6. No clear and relevant policy implications can be got from these findings.

Response: Thanks to your suggestion, we have provided the important "implications on policies or practices" of this study in the last paragraph of the discussion part in the revised manuscript.

In China, most of the work which is related to the primary or community health needs to be done by registered public health workers. Especially in the north of China which is less developed, where PHWs are in short supply and turnovers due to their low job satisfaction occur frequently⁴⁵. Meanwhile, the serious COVID-19 pandemic has largely increased the infection risk, high workload and occupational stress of public health workers, further weakening their job satisfaction. Therefore, the China National Health Commission has also put forward new requirements for stabilizing primary PHWs enhancing sense of security⁴⁶.

Based on the JD-R model and the results of this study, it is recommended that health policymakers should pay attention to the dynamic assessment of employees' thriving at work and organizational commitment when developing management systems to improve primary PHWs' job satisfaction⁴⁷. The adverse factors affecting primary PHWs' job satisfaction should be analyzed regularly, and targeted interventions should be taken. Interventions can be based on two main entry points: reducing occupational stress and risk, improving promotion pathways and salary systems. The first point is to emphasize the value of primary PHWs' work, enhance the organization's support for employees, provide them with the necessary epidemic protection supplies and personal insurance, optimize the workforce, and reduce the workload and stress of employees so that they can better cope with the challenges and stresses of their daily work⁴⁸.

The second point is to improve the remuneration of primary PHWs, and at the same time, and actively provide them with education and training, as well as a platform for career development. Meanwhile, by truly understanding the needs of this group and implementing flexible management and humanistic care, we can help elevate the motivation of primary PHWs and enhance the capacity of primary units to provide medical services⁴⁹.

VERSION 2 – REVIEW

REVIEWER	Donald Pathman University of North Carolina at Chapel Hill School of Medicine, Cecil G. Sheps Center for Health Services Research
REVIEW RETURNED	11-Mar-2022

GENERAL COMMENTS	The revised manuscript is much improved. Much of the information the reader needs is now included, and the study's foci, importance and findings are now clearer and stronger. Thank you for attending to the reviewers' many questions and suggestions. I recognize that these changes required a great amount of time. Thank you for clarifying many of the concepts within the Introduction. This helps me much better understand the meaning, importance, and relationships among the concepts, which helps me better understand the focus and importance of this study. The English of this revised draft is significantly improved but still rough in many places, with use of wrong words, incomplete sentences, and problems with grammar. In the revised "strengths and limitations of this study," appearing after the abstract, it is not clear if the statements in the first, second,
--

	and fifth bulleted points are strengths or limitations, and why. Please spell out and define in the first paragraph of Introduction what a “primary PHW” is. PHW is never clearly defined as a “public health worker,” so I and evidently the second reviewer both have been misled by the word “primary” to believe that this study might include primary care clinicians, e.g., family physicians and nurses. In the paragraph that begins, “It is known that job satisfaction, thriving at work and organizational commitment are the key factors which would affect employee’s career stability that the individual has a stable job for a certain period of time and does not have the idea of leaving easily.” I believe that “career stability” in this sentence should be “job stability.” Job stability, at least within its use in the US and western Europe, means remaining at a specific job, which is what this sentence is speaking to. Career stability refers to remaining in one’s present line of work, e.g., working as a CHW or as a physician. One can change jobs (job instability) but continue to work as a CHW (career stability). Relatedly, shouldn’t “career sustainability” appearing in the Discussion prior to the section on “implications on policies or practices” be “job sustainability” or “job retention?” Last paragraph of the Introduction: “SEM” needs to be spelled out on first usage. First paragraph of Methods. “A convenience sampling method was used to survey 601 primary PHWs in 20 primary care units (community health service centers or rural health centers) in northern provinces of China . . .” The reader still needs more clarity on how the 20 primary health care units were chosen and then how the convenience sample of PHWs was selected from among the 20 units. Thank you for providing a bit more description of the process within the responses to reviewers, but that explanation is still not adequate (for example, on what basis did “every provincial health care commission recommend primary care units? Were these the units they felt were functioned—a potential source of bias?) and this clear explanation needs to be included in the text. Thank you for attempting to reconcile references to both “601” and “650” primary PHWs being surveyed, but the current text is still not quite right. In the first paragraph of the Methods, the second sentence should indicate that 650 primary PHWs were surveyed (not 601). And in section 3.4 of Results the first sentence should read “The results showed a mean score of thriving at work of the 601 responding primary PHWs . . .”, not “601 surveyed primary PHWs.” In the “study limitations” section, the statement “there was no strong causality” should instead read something like “the relationships found in this study do not necessarily reflect causal associations.”
--	---

REVIEWER	Walter Sermeus Catholic University Leuven, Centre for Health Services and Nursing Research
REVIEW RETURNED	10-Apr-2022

GENERAL COMMENTS	The authors have well incorporated the feedback and comments from the evaluators. But still a few comments and questions remain open. First on the methods:
--

	The sampling is well described in the responses to the reviewers and should be incorporated in the article: "This study used a convenience sampling method to select the targeting participants (registered public health workers) in primary care units in the northern provinces of China, spread across Shandong, Hebei, and Henan and Shanxi provinces in China. The method of determining the units was that we randomly selected 20 from the list with hundreds of units recommended by every provincial health care commission for the survey, after obtaining their written consent." But still not clear: - convenience sample or random? Are all primary care units in the three provinces listed (targeted population), out of which randomly (?) 20 PHW were selected - stratified? Was this process repeated for each of the provinces? Were the number of units selected, weighted by the size of the province. Some more clarification is needed. A second clarification that should be made in the manuscript is the description of PHW. It is a discipline that is not commonly understood internationally (compared to physicians and nurses) and should be explained here: "Only the registered public health workers were investigated in this study. In China, public health workers are another group of healthcare provider, different with the other professional group, like doctors, pharmacists and nurses. They are mainly responsible for epidemiological investigation, assisting personnel home-isolation, community decontamination, and nucleic acid collection in the community or other primary organization during the pandemic of COVID-19." also the professional levels should be explained in more detail as they don't exist internationally: "Professional title refers to the professional level and position of registered public health workers, which is a comprehensive reflection of their work experience and work ability, and is recognized by the Ministry of Human Resources and Social Security of China. In China, this professional title system has 5 levels from junior to senior: level 1 and 2 are junior, level 3 is intermediate, and level 4 and 5 are senior titles." Results: p.38 please show how you calculate the mediated effect size of 48.7% of the total effect size (to explain for the reader). Study limitations should be discussed more extensively. It has to do with the sampling: is this sample representative for China? Why (not)? It has to do with the target group of PHW that is quite specific for China. It has to do with covid-19 in which China followed a complete different approach than other countries. English An English revision of the manuscript is still recommended in relation to some words, tense, singular/plural etc..
--	---

VERSION 2 – AUTHOR RESPONSE

Reviewer: 1

Comments to the Author:

The revised manuscript is much improved. Much of the information the reader needs is now included, and the study's foci, importance and findings are now clearer and stronger. Thank you for attending to the reviewers' many questions and suggestions. I recognize that these changes required a great amount of time.

Thank you for clarifying many of the concepts within the Introduction. This helps me much better understand the meaning, importance, and relationships among the concepts, which helps me better understand the focus and importance of this study.

The English of this revised draft is significantly improved but still rough in many places, with use of wrong words, incomplete sentences, and problems with grammar.

Response: Thank you for your valuable suggestions. We have corrected all wrong places, including inappropriate words, incomplete sentences and grammar in the revised manuscript, and the revised version was edited for proper English language, grammar, punctuation, spelling, and overall style by two highly qualified native English-speaking editors who received high-quality research training, to make it meet the requirements for acceptance by BMJ Open.

In the revised "strengths and limitations of this study," appearing after the abstract, it is not clear if the statements in the first, second, and fifth bulleted points are strengths or limitations, and why.

Response: The first and second bulleted points are STRENGTHS and the fifth point is LIMITATION. In view of your suggestion, I have added more content to these points so that the reader can clearly know whether the points are strengths or limitations.

Please spell out and define in the first paragraph of Introduction what a "primary PHW" is. PHW is never clearly defined as a "public health worker," so I and evidently the second reviewer both have been misled by the word "primary" to believe that this study might include primary care clinicians, e.g., family physicians and nurses.

Response: Thanks to your suggestion, we have placed the definition of PHWs in the second paragraph of the Introduction section so that the reader can understand the characteristics and job responsibilities of this group of healthcare workers with Chinese characteristics.

In the paragraph that begins, “It is known that job satisfaction, thriving at work and organizational commitment are the key factors which would affect employee’s career stability that the individual has a stable job for a certain period of time and does not have the idea of leaving easily.” I believe that “career stability” in this sentence should be “job stability.” Job stability, at least within its use in the US and western Europe, means remaining at a specific job, which is what this sentence is speaking to. Career stability refers to remaining in one’s present line of work, e.g., working as a CHW or as a physician. One can change jobs (job instability) but continue to work as a CHW (career stability). Relatedly, shouldn’t “career sustainability” appearing in the Discussion prior to the section on “implications on policies or practices” be “job sustainability” or “job retention?”

Response: Thank you very much for your good suggestions, and I was especially touched by the large paragraph you wrote telling us we should use more appropriate words. Your rigor and kindness impressed our team members and made us an example to follow. Thank you for your passion for research!

Last paragraph of the Introduction: “SEM” needs to be spelled out on first usage.

Response: Thank you for your suggestion, it has been revised into structural equation model (SEM).

First paragraph of Methods. “A convenience sampling method was used to survey 601 primary PHWs in 20 primary care units (community health service centers or rural health centers) in northern provinces of China . . .” The reader still needs more clarity on how the 20 primary health care units were chosen and then how the convenience sample of PHWs was selected from among the 20 units. Thank you for providing a bit more description of the process within the responses to reviewers, but that explanation is still not adequate (for example, on what basis did “every provincial health care commission recommend primary care units? Were these the units they felt were functioned—a potential source of bias?) and this clear explanation needs to be included in the text.

Response: This study used convenience sampling to determine the selection of primary health care units, and the random number table used to determine the targeting participants. We surveyed 20 primary care units in northern provinces of China with whom we had a partnership (convenience sampling), and then used the random number table method to identify a total of 650 primary PHWs, the response rate was 92.46%, which resulted in a sample of 601 public health workers. The sample size was not determined according to the number of primary units in each province according to the weighting method by poor consideration at that time. Therefore, we added it to the study’s limitation that in the sampling process may be the potential source of bias.

Thank you for attempting to reconcile references to both “601” and “650” primary PHWs being surveyed, but the current text is still not quite right. In the first paragraph of the Methods, the second sentence should indicate that 650 primary PHWs were surveyed (not 601). And in section 3.4 of Results the first sentence should read “The results showed a mean score of thriving at work of the 601 responding primary PHWs . . .”, not “601 surveyed primary PHWs.”

Response: In this study, 650 PHWs from 20 primary healthcare units in some provinces in northern China participated in the survey, and 601 responding participations completely filled in the relevant scales, resulting in 601 data finally. Thank you for pointing out such mistake I made, and your explanations make me understand the difference between “surveyed participants” and “responding participants” two terms. We have completed the corresponding changes in the revised manuscript.

In the “study limitations” section, the statement “there was no strong causality” should instead read something like “the relationships found in this study do not necessarily reflect causal associations.”

Response: Thank you for your suggestion, we have revised this original sentence into “The relationships among variables found in this study do not necessarily reflect causality” in the manuscript.

Reviewer: 2

Comments to the Author:

The authors have well incorporated the feedback and comments from the evaluators. But still a few comments and questions remain open.

First on the methods:

The sampling is well described in the responses to the reviewers and should be incorporated in the article: "This study used a convenience sampling method to select the targeting participants (registered public health workers) in primary care units in the northern provinces of China, spread across Shandong, Hebei, and Henan and Shanxi provinces in China. The method of determining the units was that we randomly selected 20 from the list with hundreds of units recommended by every provincial healthcare commission for the survey, after obtaining their written consent."

But still not clear: - convenience sample or random? Are all primary care units in the three provinces listed (targeted population), out of which randomly (?) 20 PHW were selected- stratified? Was this

process repeated for each of the provinces? Were the number of units selected, weighted by the size of the province. Some more clarification is needed.

Response: This study used convenience sampling to determine the selection of primary health care units, and the random number table used to determine the targeting participants. We surveyed 20 primary care units in northern provinces of China with whom we had a partnership (convenience sampling), and then used the random number table method to identify a total of 650 primary PHWs, the response rate was 92.46%, which resulted in a sample of 601 public health workers. The sample size was not determined according to the number of primary units in each province according to the weighting method by poor consideration at that time. Therefore, we added it to the study's limitation that in the sampling process may be the potential source of bias.

A second clarification that should be made in the manuscript is the description of PHW. It is a discipline that is not commonly understood internationally (compared to physicians and nurses) and should be explained here: "Only the registered public health workers were investigated in this study. In China, public health workers are a separate group of healthcare provider, different with the other professional group, like physicians, pharmacists and nurses. They are mainly responsible for epidemiological investigation, quarantine management of patients with infectious diseases, environmental disinfection, and specimen collection in the community or other primary organization."

Response: Thanks to your suggestion, we have placed the definition of PHWs in the second paragraph of the **Introduction** section so that the reader can understand the characteristics and responsibilities of the PHWs, the indispensable part of healthcare workers with Chinese characteristics.

Also the professional levels should be explained in more detail as they don't exist internationally: "Professional title refers to the professional level and position of registered public health workers, which is a comprehensive reflection of their work experience and work ability, and is recognized by the Ministry of Human Resources and Social Security of China. In China, this professional title system has 5 levels from junior to senior: level 1 and 2 are junior, level 3 is intermediate, and level 4 and 5 are senior titles."

Response: Thanks to your suggestion, we have put the explanation of 'professional title' to in Table 1 of the revised manuscript. Professional title is a comprehensive reflection of the registered public health workers' work ability and work position in the related field. China national Health Commission uses 'professional title' to classify the competencies of public health workers.

Results:

p.38 please show how you calculate the mediated effect size of 48.7% of the total effect size (to explain for the reader).

Response: Thanks to your suggestion, we have added a description of the calculation of the mediating effect value in the revised manuscript. It is the value of the indirect effect divided by the value of the total effect to get the percentage of the mediating effect.

Study limitations should be discussed more extensively. It has to do with the sampling: is this sample representative for China? Why (not)? It has to do with the target group of PHW that is quite specific for China. It has to do with covid-19 in which China followed a completely different approach than other countries.

Response: First of all, as you referred in the comments, the target group of public health workers that is quite specific for China, so this sample population is undoubtedly representative for the primary PHWs of China. Unlike China, the healthcare providers involved in the front-line fight against the COVID-19 pandemic in other countries are mainly physicians and nurses. So, there is a limitation in the generalization of the results. Although it may seem like a different group with PHWs, this study still has implications for other countries, such as healthcare policymaker should recognize the importance of thriving and organizational commitment of workers providing medical and nursing care to patients with COVID-19, and use relevant measures to motivate staff thriving and thus improve job satisfaction as a way to face the greater challenges.

An English revision of the manuscript is still recommended in relation to some words, tense, singular/plural etc.

Response: Thank you for your valuable suggestions. We have corrected all wrong places, including inappropriate words, incomplete sentences and grammar in the revised manuscript, and the revised version was edited for proper English language, grammar, punctuation, spelling, and overall style by two highly qualified native English-speaking editors who received high-quality research training, to make it meet the requirements for acceptance by BMJ Open.